# Ganglion Cell Layer Thinning in Alzheimer’s Disease

**DOI:** 10.3390/medicina56100553

**Published:** 2020-10-21

**Authors:** Alicia López-de-Eguileta, Andrea Cerveró, Ainara Ruiz de Sabando, Pascual Sánchez-Juan, Alfonso Casado

**Affiliations:** 1Department of Ophthalmology, ‘Marqués de Valdecilla’ University Hospital, University of Cantabria, Institute for Research ‘Marqués de Valdecilla’ (IDIVAL), 39008 Santander, Spain; alicialeguileta@gmail.com (A.L.-d.-E.); andreacervero27@gmail.com (A.C.); 2Genetic deoartment, Navarrabiomed, 31008 Pamplona, Spain; ainararse@hotmail.com; 3Neurology Department and Centro de Investigación Biomédica en Red sobre Enfermedades Neurodegenerativas (CIBERNED), ‘Marqués de Valdecilla’ University Hospital, University of Cantabria, Institute for Research ‘Marqués de Valdecilla’ (IDIVAL), 39008 Santander, Spain; pascualjesus.sanchez@scsalud.es

**Keywords:** ganglion cells, optical coherence tomography, Alzheimer’s disease, mild cognitive impairment

## Abstract

The main advantages of optical retinal imaging may allow researchers to achieve deeper analysis of retinal ganglion cells (GC) in vivo using optical coherence tomography (OCT). Using this device to elucidate the impact of Alzheimer’s disease (AD) on retinal health with the aim to identify a new AD biomarker, a large amount of studies has analyzed GC in different stages of the disease. Our review highlights recent knowledge into measuring retinal morphology in AD making distinctive between whether those studies included patients with clinical dementia stage or also mild cognitive impairment (MCI), which selection criteria were applied to diagnosed patients included, and which device of OCT was employed. Despite several differences, previous works found a significant thinning of GC layer in patients with AD and MCI. In the long term, an important future direction is to achieve a specific ocular biomarker with enough sensitivity to reveal preclinical AD disorder and to monitor progression.

## 1. Introduction

Optical coherence tomography (OCT) is the most widely used imaging device in ophthalmic clinical practice [1]. This noninvasive, fast, and inexpensive technology employs retroreflected light to achieved cross-sectional structure images of the retina and the anterior eye chamber with high resolution. OCT imaging reveals individual neuronal layers of the retina, including ganglion cell layer (GCL) [2]. Initially, the main utility of GCL assessment was the diagnosis and treatment of ocular diseases such as glaucoma [2,3]. There is growing evidence to incorporate OCT imaging into clinical diagnosis managing neurodegenerative diseases, including Alzheimer’s disease (AD), Parkinson’s disease, and multiple sclerosis [4,5,6,7]. The aim of this review is to analyze the use of ganglion cell layer measurement in AD.

AD is a the most prevalent neurodegenerative disorder and the leading cause of dementia in the elderly [8]. Definitive diagnosis of AD is given by pathological features like intracellular neurofibrillary tangles of hyperphosphorylated tau protein (*p*-Tau) and extracellular beta amyloid (Aβ) protein deposits throughout the brain [9,10]. These well-known neuropathological hallmarks of AD initiated decades before it is clinically expressed, where there might be a window to treat AD [11,12]. Current diagnostic modalities for AD biomarkers are restricted by high costs and limited availability such as the use of magnetic resonance imaging (MRI) or positron emission tomography (PET), as well as standardization problems and invasiveness of cerebrospinal fluid (CSF) biomarkers), or suboptimal specificity and sensitivity (genetic markers, serum amyloid) [13,14,15]. These limitations are the leading cause of the investigation of new AD biomarkers involving the evaluation of the eye.

It has long been demonstrated that patients with early AD suffer impairments in visual acuity [16], contrast sensitivity [17], color perception [18], visual field [19], and motion perception [20]. Initially, visual disorders in AD were thought to be exclusively due to parietal and primary visual cortex pathology. However, increasing evidence showed that the anterior visual pathway degeneration also plays a role in AD pathology. As previously stated, OCT is able to provide a morphological assessment of the retinal layers and optic nerve structures, and several studies have been performed to assess differences between AD and control patients. GCL reduction might be the most common finding in the literature, although the assessment of this retinal layer is recently feasible.

## 2. Material and Methods

A systematic review was performed. The authors reviewed the literature using PubMed. The online citation index service PubMed was searched using the keywords optical coherence tomography and mild cognitive impairment or Alzheimer’s disease. Manuscripts including those keywords with available OCT technology published in peer-reviewed publications were considered for this review.

We identified 24 eligible studies, involving 808 probable AD patients, all of which were cross-sectional studies. Neither retrospective meta-analyses nor OCT angiography studies were included in this review. In original research articles, the revisions considered the structures of the retina investigated, the significance of the results, the use of AD biomarkers, which OCT device was employed, the design of the study, demographics, groups sizes and number of eyes included, and the characteristics of the different groups of the studies.

## 3. Results

OCT constitutes an important advancement and powerful tool to evaluate alterations of the optic nerve and the retina and provides an opportunity for objective quantitative measurements and in vivo real-time images of ocular structures related with neurological diseases. This review included 24 most important AD and OCT studies that focused in retinal GCL in order to present clear results easy to be understood. As it could be depicted in Table 1, most of these studies found a significant thinning of the retinal nerve fiber layer (RNFL) and GCL between probable AD patients and healthy controls (HC), using both Cirrus and Spectralis HD-OCT. Figure 1 illustrates the structure of the retina that is analyzed with OCT. Different densities of nuclear layers due to neuron bodies are reflected in OCT images and allows these devices to perform the layers’ segmentation.

Currently, two main OCT devices are used in the clinical practice and they performed the retinal segmentation analysis differently, as it can be appreciated in Figure 2. Figure 2A represents the OCT imaging with Cirrus spectral domain SD-OCT (Carl Zeiss Meditec, Dublin, CA, USA), whereas Figure 2B shows the OCT imaging with Spectralis SD-OCT (Heidelberg Engineering, Heidelberg, Germany). Cirrus SD-OCT segmented GCL including the inner plexiform layer (IPL) whereas Spectralis accomplished retinal segmentation including GCL solely. Studies made using OCT Cirrus refer to GCL as ganglion cell-inner plexiform layer (GCIPL) and those using Spectralis label it as GCL. This fact elucidates why GCL and GCIPL thickness measurements should not be compared between them. Along this text and in Table 1, GCIPL or GCL are differently used according to Cirrus or Spectralis OCT, respectively. Figure 3 and Figure 4 represent OCT images of a healthy control (HC) and an AD patient, respectively, with Cirrus OCT assessment in the top of both figures and Spectralis OCT assessment in the bottom. At the top of both figures, Cirrus OCT images showed segmentation lines (“horizontal tomography”) and deviation maps. In Figure 3, no damage was demonstrated. However, in Figure 4, three and two sectors of color map can be appreciated in right and left eye, respectively. Similarly, deviation and thickness maps are exhibited also damaged. Similarly, in the bottom of both figures, Spectralis OCT exhibit a complete red circle for an HC and yellow-red colors for an AD patient. Currently, this OCT device did not have a normative database to compare GCL with normal population. For this reason, studies using Spectralis OCT must include HC. Nevertheless, Spectralis OCT yields 64 values of GCL of the entire macula.

In recent years, increasing efforts have been made to discover new biomarkers with the aim to improve AD diagnosis in early stages. Hinton et al. stated the feasible damage of the eye due to AD and provided histopathological evidence of optic neuropathy and degeneration of retinal GCL in AD subjects [21]. Some years later, postmortem studies showed that degeneration of the GCL occurs preferentially in superior and inferior sectors, as well as in the central retina, particularly in the temporal foveal region [22,23]. Lately, both Aβ and neurofibrillary tangles were detected in some parts of the visual system in probable AD patients, including the retina [24,25]. Interestingly, in a mouse model of AD, Aβ deposits were specifically in the GCL [26]. In assent with this finding, Koronyo et al. demonstrated histopathologically that GCL damage due to AD might be related with intracellular neurofibrillary tangles of *p*-Tau and extracellular Aβ protein deposits throughout the retina and not related with other etiologies of dementia [27]. Extensive loss of ganglion cells and their axons has been reported by histopathologic studies in eyes from probable AD patients and AD animal models [28]. There might be two mechanisms which explain GCL damage. The first proposed that cerebral pathologic features of AD may affect the visual pathway and cause retrograde degeneration of the optic nerve [29], and subsequently damage of the GCL, because AD pathologic features can be found in subcortical visual centers, including the lateral geniculate nucleus and superior colliculus [30]. In agreement with this hypothesis, GC abnormalities also were associated with non-AD dementias [31,32,33], strokes [34,35], and other neurodegenerative diseases including multiple sclerosis [36,37,38], neuromyelitis optica [37], and cerebral atrophy [39]. Alternatively, it is also possible that AD pathologic features occur simultaneously both in the brain and the retina, leading to thinning of the retinal neuronal layers. Aβ pathologic characteristics, including Aβplaques and specific signs of neuroinflammation, have been identified in ocular tissues of both probable AD patients [24,40] and animal models of AD [27,41,42,43,44,45].

Several studies reported retinal and optic nerve changes in patients with AD using OCT imaging in vivo, generating interest in the use of these parameters as biomarkers for early detection of AD [46,47,48]. Retinal changes may be an early event in the course of AD, and retinal OCT may provide insights for assessing neurodegeneration in the brain [29]. As previously stated, OCT is a reliable noninvasive, transpupillary technique that provides high-resolution cross-sectional images of RNFL and macular volume [49,50,51,52]. RNFL thickness is believed to inform about axonal loss, whereas macular volume reflects neuronal loss of GCL, since the neuronal bodies and dendrites are located in the GCL, mostly in the macula [49].

Initially, evidence of total macular thickness decreased in patients with AD was demonstrated with time domain OCT (TD-OCT) [53] and stratus OCT [49,54]. Subsequently, it was confirmed by several independent groups using modern OCT devices, such as spectral-domain OCT [55,56,57,58,59].

Spectral-domain OCT (SD-OCT), a Fourier domain OCT technique, provided dramatically increased scanning speed and higher axial resolution when compared to TD-OCT technology allowing the study of GCL and analysis of the GCIPL layers [52,60]. The classical site of GCIPL measurement in the studies is macula lutea, where GCL has more than one cell layer [53]. As a consequence of macular segmentation program development, Marziani et al. reported significant reductions in combined RNFL and GCL thickness (RNFL + GCL + IPL) in the macular region [60]. However, the authors were not able to determinate which layer was most affected by AD due to poor segmentation OCT system. Lately, a number of studies suggested including RNFL in the GCIPL analysis in the macular area may influence the sensitivity for revealing GCL damage, so they measured GCIPL without including the RNFL, and found significant GCIPL thinning in AD [52,61,62,63,64,65,66,67,68,69] and MCI patients compared to HC [61,64].

On the contrary, some in vivo studies displayed controversial GCL measurements using SD-OCT reporting that RNFL and GCL thickness might not be unable to distinguish AD dementia from MCI and normal controls in clinically well-characterized series [65,66]. The authors themselves hypothesized that a larger series would be necessary to delineate significant differences between the groups studied. In our opinion, their study has methodological limitations. Although PET imaging was performed as an inclusion criterion for AD, neither the ligand used nor the imaging result are detailed. Furthermore, patients with glaucoma were excluded, but the criteria for exclusion are not clearly or correctly described.

Another controversial study is from Ferrari et al. [69]. They described significant group differences regarding GCIPL, being reduced in moderate AD versus mild AD and versus MCI. However, nonsignificant GCIPL thinning was detected in MCI compared with HC. One explanation may be that they explored GCIPL in the peripapillary area, and this is not ideal site to detect early GCL loss due to the poor representation of ganglion cells at this location. Macular GCIPL thinning may be a more sensitive marker of earlier neurodegeneration in MCI and AD than evaluation of the RNFL.

Recently, our group have published an investigation about retinal damage in AD assessed by Spectralis OCT, reporting promising results. The study included highly characterized patients with detailed neurocognitive testing and positive to ^11^C-labeled Pittsburgh Compound-B with positron emission tomography [64] analysis that could readily differentiate between participants with normal cognition from dementia due to AD. AD and MCI patients were recruited and compared among them and HC. The investigation reported a significant thinning of RNFL, GCL, IPL, and outer nuclear layer (ONL). Interestingly, temporal sector of GCL showed the greatest area under the curve value.

Aforementioned studies have some design limitations. One significant gap could be that the thinning of GCL might be due to other eye conditions such as glaucoma, arteritic or nonarteritic optic neuropathy, or other neurological disease. For this reason, results of GCL thinning might be used to understand the pathophysiology of AD, but they should be carefully interpreted. Future techniques might provide more specific information about retinal ganglion cell degeneration in AD. Besides, definition of MCI was used meeting research diagnostic criteria for probable AD MCI or with evidence of the AD pathophysiological process (in most cases defined by a positive amyloid-PET) following the recommendations of the National Institute on Aging-Alzheimer’s Association [70], but criteria should be consistent. In addition, all studies screened the eyes with the OCT technology after pupil dilation, except one [64], which might constitute a limitation. Finally, it is important to focus that his review revealed that limited research had focused exclusively on screening the eyes of study subjects with and with no cognitive decline using optical coherence tomography, neuropsychological tests, and in vivo neuroimaging techniques. Besides, a few studies used MRI, PET, or AD biomarkers for the diagnosis in the AD group.

## 4. Conclusions

The investigation of eye biomarkers in AD using OCT assessment remains an area of active research. Reviewing literature about this subject, it is feasible to find a large amount of studies, and several of them show significant damage of the GCL or GCIPL in probable AD patients, even during the early stage of the disease and using different OCT devices such as Cirrus and Spectralis. We consider these results might provide more detailed information about the physiopathology of AD and the relevance of GCL in neurological diseases. In terms of diagnosis, current available techniques show temporal region thinning of GCL, which might be the most reliable indicator of a possible MCI or AD patient. It is necessary to be aware that this thinning could be consequence of other ocular or neurological circumstance. However, today might be soon to consider optic nerve or retinal biomarkers as reliable biomarkers of AD, as a limited number of studies have compared OCT retinal measures with neuroimaging biomarkers and a very few longitudinal within-subject studies of retinal structural changes in AD are published.

## Figures and Tables

**Figure 1 medicina-56-00553-f001:**
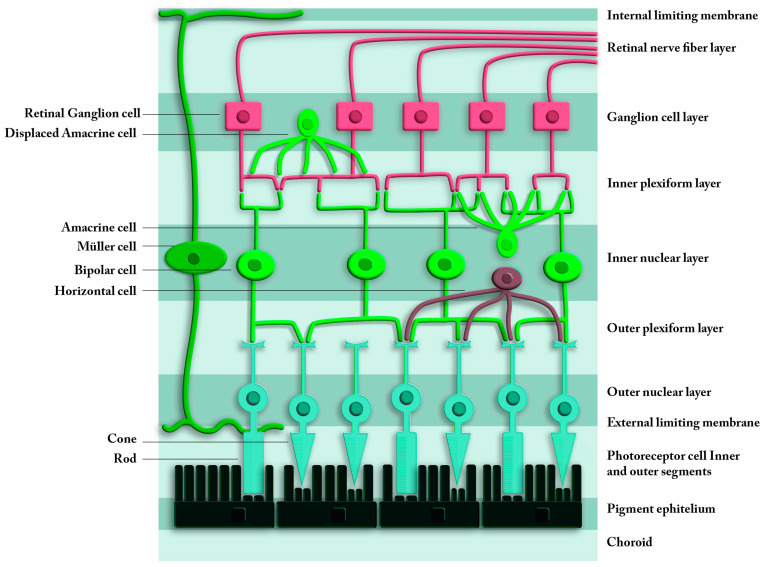
Schematic view of the retina. The upper layers are the innermost layers, in contact with vitreous humor. The lower layers of the scheme are those that are in contact with the choroid. From the innermost to the outermost, the layers of the retina are: retinal nerve fiber layer, ganglion cell layer, inner plexiform layer, inner nuclear layer, outer plexiform layer, outer nuclear layer, external limiting membrane, and photoreceptor layer.

**Figure 2 medicina-56-00553-f002:**
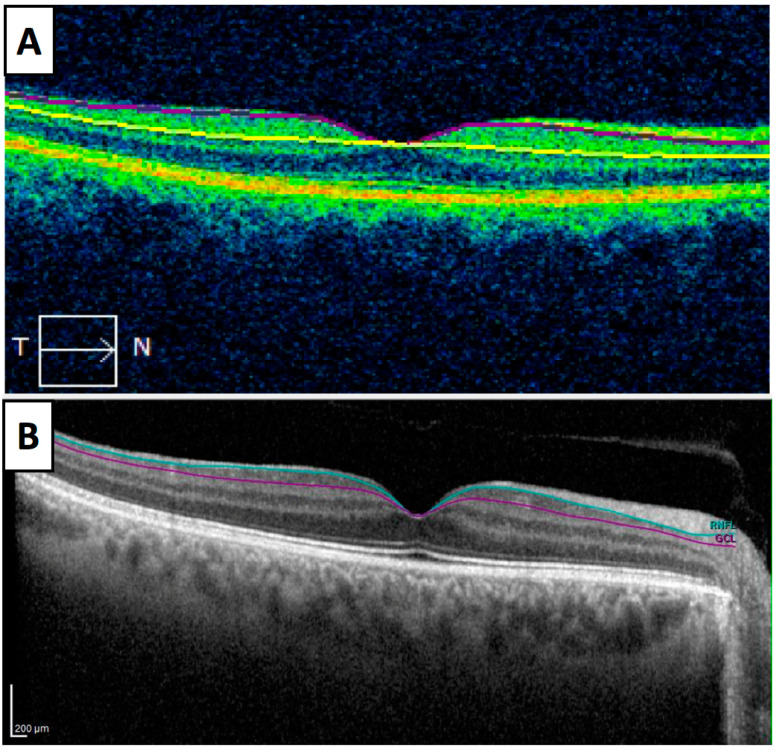
Horizontal scans of Cirrus optical coherence tomography (OCT) (**A**) and Spectralis OCT (**B**). (**A**) Segmentation of ganglion cell layer (GCL) with Cirrus OCT includes GCL and inner plexiform layer, GCIPL, between purple and yellow lines. (**B**) Segmentation of GCL using Spectralis OCT includes GCL in an exclusive manner, between blue and purple lines.

**Figure 3 medicina-56-00553-f003:**
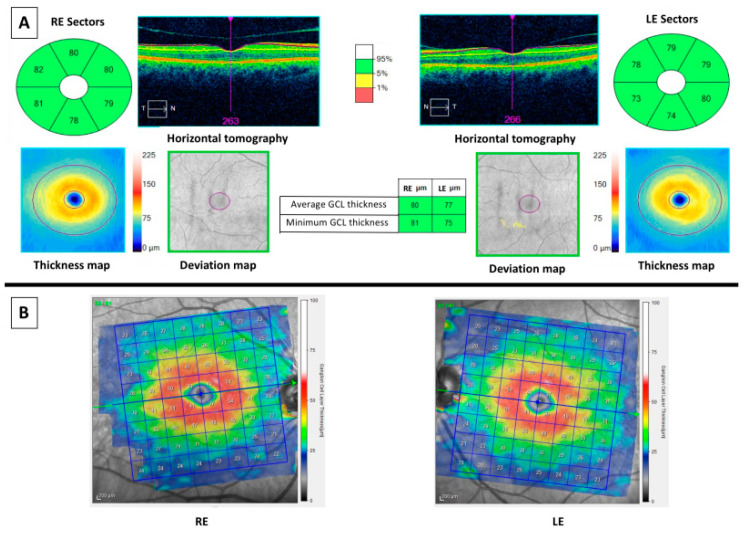
OCT results of a 67 years-old male healthy control. At left: images of the right eye. At right: images of the left eye. (**A**) Cirrus optical coherence tomography (OCT) results. Top middle (horizontal tomography): images of horizontal scans to confirm correct segmentation of ganglion cell-inner plexiform layer (GCIPL). Green color values (in microns): sectors of GCIPL compared with normative database. The thickness map shows the thickness in a color map (the caption of the colors is at right of the maps), whereas the deviation map shows yellow or red color if a pixel of GCIPL is low of fifth or first percentile, respectively. Middle columns showed the average and the minimum values (in microns) of GCIPL of both eyes, colored in green if they are thicker than fifth percentile. (**B**) Spectralis OCT showed a color map (the caption of the colors is at right of the maps) and values of thicknesses (in microns) of GCL.

**Figure 4 medicina-56-00553-f004:**
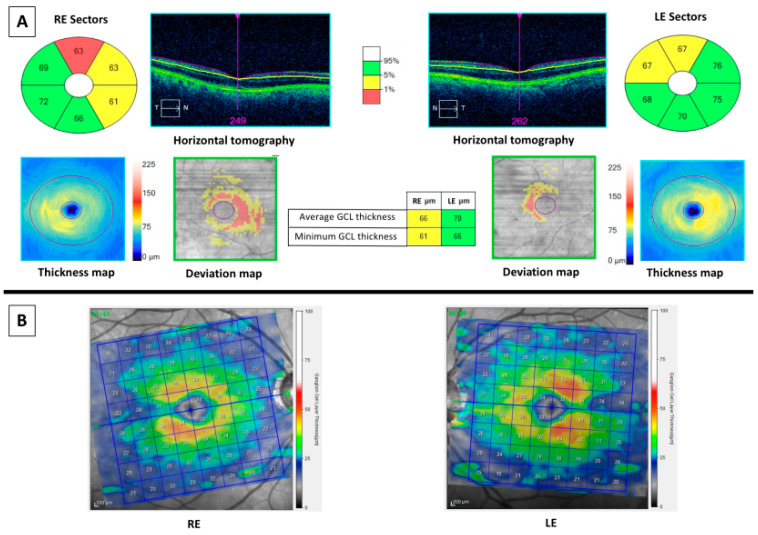
OCT results of a 67 years-old male MCI patient. At left: images of the right eye (RE). At right: images of the left eye (LE). (**A**) Cirrus optical coherence tomography (OCT) results. Top middle (horizontal tomography): images of horizontal scans to confirm correct segmentation of ganglion cell-inner plexiform layer (GCIPL). Six sectors color values: sectors of GCIPL compared with normative database. Three and two sectors were thinner than fifth percentile in RE and LE, respectively. The thickness map shows the thickness in a color map (the caption of the colors is at right of the map), whereas the deviation map shows yellow or red color if a pixel of GCIPL is low of fifth or first percentile, respectively. Middle columns showed the average and the minimum values of GCIPL of both eyes, colored with yellow for the RE as they are thinner than fifth percentile. (**B**) Spectralis OCT showed a color map (the caption of the colors is at right of the maps) and values of thicknesses (in microns) of GCL. Comparing with Figure 3B, where a red ring could be appreciated, the RE of this patient is not red colored and LE is little red colored.

**Table 1 medicina-56-00553-t001:** Studies evaluating retinal biomarkers in Alzheimer’s disease using optical coherence tomography. Optical coherence tomography (OCT), ganglion cell layer (GCL), Alzheimer’s disease (AD), mild cognitive impairment (MCI), subjective memory complaints (SMC), healthy controls (HC), normotensive glaucoma (NTG), retinal nerve fiber layer (RNFL), ganglion cell layer (GCL), ganglion cell inner plexiform layer (GCIPL), Bruch’s membrane opening-minimum rim width (BMO-MRW), inner plexiform layer (IPL), outer nuclear layer (ONL), inner nuclear layer (INL), outer plexiform layer (OPL), lamina cribrosa (LC), magnetic resonance imaging (MRI), positron emission tomography (PET), computed tomography (CT). Significant results are showed as (%, *p*).

Source	OCT Exam: Layers	Macular or GCL Results	AD Biomarkers	OCT Platform	Cross-Sectional	Subjects	Sample Size (Eyes)
Iseri et al. 2006	RNFL and macula	Thinner (23%, *p* < 0.001)	No	Zeiss Stratus	Yes	AD HC	AD 28 eyes (*n* = 14) HC 30 eyes (*n* = 15) Age-matched
Moschos et al. 2012	RNFL and macula	Thinner (7%, *p* = 0.034)	No	Zeiss Stratus	Yes	AD HC	AD (*n* = 30) HCs (*n* = 30) Age and sex matched
Marziani et al. 2013	RNFL + GCL combined	Thinner (12.8%, *p* = 0.008)	No	RTVue-100 and Heidelberg Spectralis	Yes	AD HC	AD (*n* = 21) HC (*n* = 21) Age-matched
Garcia-Martin et al. 2014	RNFL and macula	Mild AD had a significant decrease in RNFL (9.24%, *p* = 0.015), of some macular regions and in the total macular volume (9.34%, *p* = 0.024).	No	Topcon 3D OCT-100	Yes	Mild AD HC	Mild AD (*n* = 20) HC (*n* = 28) Age-matched
Ascaso et al. 2014	RNFL and macula	RNFL was thinner in -MCI vs. HC (8.5%, *p* = 0.001)-AD vs. MCI (24.8%, *p* = 0.001) -AD vs. HC (37.5%, *p* = 0.001) Macular volume in mm^3^: -HC had greater macular volume vs. AD (12.4%, *p* = 0.001)	No	Zeiss Stratus	Yes	AD MCI HC	AD (*n* = 18) MCI (*n* = 21) HC (*n* = 41)
Eraslan et al. 2015	RNFL and GCL	-RNFL Thinner in AD and NTG vs. HC (8%, *p* = 0.004). -GCL (8.8%, *p* = 0.001) -No difference between AD and NTG.	No	RTVue-100	Yes	NTG AD HC	NTG (*n* = 18) AD (*n* = 20) HC (*n* = 20)
Bayhan et al. 2015	GCL and choroid	Reduced choroidal (12.1%, *p* = 0.01) and macular GCL (5.9%, *p* = 0.001) thicknesses in AD	CT or MRI	Zeiss Stratus	Yes	AD HC	AD (*n* = 31) HC (*n* = 30) Age matched
Cheung et al. 2015	RNFL and GCIPL	- AD had GCIPL thinning in all sectors (AVG 5.4%, *p* = 0.039) and RNFL in Superior quadrant vs. HC (6.5%, *p* = 0.001) -GCIPL reduction in MCI (5.1%, *p* = 0.009)	CT or MRI	Zeiss Cirrus	Yes	MCI AD HC	AD (*n* = 100) MCI (*n* = 41) HC (*n* = 123)
Pillai et al. 2016	RNFL, macula GCL	No differences (*p* = 0.35 and *p* = 0.17)	MRI	Zeiss Cirrus	Yes	AD MCI No AD Dementia Parkinson HC	AD (*n* = 21) MCI (*n* = 21) no AD dementia (*n* = 20) PD (*n* = 20) HC (*n* = 34) Age-/sex-matched
Garcia Martin et al. 2016	RNFL, GCL, INL, IPL, ONL, OPL	Thinner RNFL (5.6%, *p* = 0.004), GCL (2.8%, *p* = 0.04) and IPL (2.3%, *p* = 0.018)	No	Heidelberg Spectralis	Yes	AD HC	AD (*n* = 150) HC (*n* = 75) Age-matched
Liu et al. 2016	GCIPL	Thinner (2.1%, *p* = 0.003)	Yes. MRI	Zeiss Cirrus	Yes	MCI AD HC	MCI (*n* = 68) AD (*n* = 47) HC (*n* = 65)
Choi et al. 2016	RNFL andGCIPL	-RNFL thinner in temporal sector (14.9%, *p* = 0.04). -GCIPL thinner in inferior sector (14.5%, *p* = 0.004).	Yes	Zeiss cirrus	Yes	MCI AD HC	AD (*n* = 42) MCI (*n* = 26) HC (*n* = 66) Age-matched, age as a covariate
Gimenéz Castejon et al. 2016	Macula	Macular thickness reduction in MCI (5.7%, *p* = 0.05) vs. HC and in SMC vs. HC (4.9%, *p* = 0.05)	No	Zeiss cirrus	Yes	SMC MCI HC	SMC *n* = 24 MCI *n* = 33 HC *n* = 25
Snyder et al. 2016	IPL	Thicker (5.8%, *p* = 0.029)	Yes (florbetapir PET imaging)	Heidelberg Spectralis	Yes	SMC	SMC (*n* = 63) Age-matched, age as a covariate
Kwon et al. 2017	RNFL and macula	RNFL average thinner in AD vs. MCI (7.8%, *p* = 0.011). Macular thickness was thinner from HC to MCI and to AD, but no significant.	Yes (MRI)	Zeiss Cirrus	Yes	Gender and race unknown	AD (*n* = 15) MCI (*n* = 15) HC (*n* = 15)
Ferrari et al. 2017	RNFL and GCIPL	Thinning (6.4%, *p* = 0.023) (15.9%, *p* = 0.009)	No	Heidelberg Spectralis	Yes	MCI AD HC	AD (*n* = 39) MCI (*n* = 27) HC (*n* = 49) Age-matched, age as a covariate
Golzan et al. 2017	RNFL and GCL	GCL thinner (5.2%, *p* = 0.02) No RNFL differences	Yes (MRI, florbetapir PET imaging)	Heidelberg Spectralis	Yes	AD HC	AD *n* = 73 HC *n* = 28 Age-matched, age as a covariate
Poroy et al. 2018	RNFL and macula	Foveal thickness and volume were higher in AD (5.5%, *p* = 0.023). RNFL and other macular region not different.	No	Zeiss Stratus	Yes	AD HC	AD (*n* = 21) HC (*n* = 25) Age-matched
den Haan et al. 2018	RNFL and macula	No differences	Yes (MRI, PET, CSF)	Heidelberg Spectralis	Yes	AD HC	Early onset AD (*n* = 15) HC (*n* = 15)
Lad et al. 2018	RNFL, GCIP	No differences	No	Heidelberg Spectralis	Yes	MCI AD HC	MCI (*n* = 15) AD (*n* = 15) HC (*n* = 18)
Uchida et al. 2018	ONL	No differences	Yes (MRI)	Zeiss Cirrus	Yes	AD MCI non-AD Dementia HC	AD (*n* = 24) MCI (*n* = 22) non-AD dementia (*n* = 20) HC (*n* = 36)
Santos et al. 2018	RNFL, GCL, OPL, ONL, IPL, INL	RNFL volume (*p* = 0.05), OPL temporal (*p* = 0.04), ONL (*p* = 0.026) and IPL volume (*p* = 0.020) and inferior thinner over a 27-month follow-up	Yes (florbetapir PET imaging, head CT)	Heidelberg Spectralis	No, 27 months	Preclinical AD HC	Preclinical AD (*n* = 56) Age-matched
López de Eguileta et al. 2019	RNFL, GCL, BMO-MRW, IPL, ONL, LC	RNFL (2.8%, *p* = 0.004), GCL (8.7%, *p* = 0.006), IPL (5.2%, *p* = 0.011) & ONL (7.9%, *p* = 0.010) showed significant thinning in eyes of patients with positive 11C-PiB PET/CT	Yes (^11^C-labeled Pittsburgh Compound-B PET imaging, head CT)	Heidelberg Spectralis	Yes	MCI AD HC	MCI (*n* =51) AD (*n* =12) HC (*n* = 63)

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
