# Peer review of "Ganglion Cell Layer Thinning in Alzheimer’s Disease"

_medicina, 2020, doi:10.3390/medicina56100553_

Round 1

Reviewer 1 Report

In this review, the Authors investigate the ability OCT to estimate the ganglion cell loss in eyes with AD or MCI.

The study is well conducted and written. Results are precise, it sounds well and could be interesting for the readership of the journal, even if not at all new.

Perhaps, Authors could better empathize that only a limited number of studies have compared OCT retinal measures with neuroimaging biomarkers to benchmark OCT sensitivity as potential screening markers for AD.
Also, we know longitudinal within-subject studies of retinal structural changes in AD are very few.

Author Response

Perhaps, Authors could better empathize that only a limited number of studies have compared OCT retinal measures with neuroimaging biomarkers to benchmark OCT sensitivity as potential screening markers for AD.
Also, we know longitudinal within-subject studies of retinal structural changes in AD are very few.

We strongly appreciate the suggested changes by Reviewer 1, so we have added the following sentences:

“However, nowadays might be soon to consider optic nerve or retinal biomarkers as a reliable biomarkers of AD, as a limited number of studies have compared OCT retinal measures with neuroimaging biomarkers and very few longitudinal within-subject studies of retinal structural changes in AD are published.”

Reviewer 2 Report

The review article aims to explore the utility of retinal ganglion cell layer thickness as a potential biomarker for AD and its earlier stages.  It does so by reviewing studies that used OCT as a non-invasive eye imaging device to generate optical cross-section of the posterior pole of the eye that enables the segmentation of different layers of the retina in order to measure its thickness. There are several systemic reviews and meta-analysis have been published exploring the utility of OCT imaging in AD and MCI. Just to mention few:  

Recent meta-analysis also revealed the importance of GCL-IPL and GCC in AD and well discussed the issues that this manuscript highlighted and went well beyond that [1].  Another Meta-analysis highlighted that although there are retinal changes in AD, it might be too small to translate finding into clinic [2].

Further reviews:

Major review: Retinal changes in Alzheimer's disease— integrated prospects of imaging, functional and molecular advances[3]

Review: Afferent and Efferent Visual Markers of Alzheimer’s Disease: A Review and Update in Early Stage Disease[4]

The manuscript states that the major focus of the review will be the ganglion cell layer but then it talks about all the other retinal layers too. It doesn’t read well and feels chaotic. The results and the discussion section are not separated.

There is no novel information in this manuscript. Unfortunately, I do not see how the field would benefit from this manuscript.

References:  

[1]          Chan VTT, Sun Z, Tang S, Chen LJ, Wong A, Tham CC, Wong TY, Chen C, Ikram MK, Whitson HE, Lad EM, Mok VCT, Cheung CY (2018) Spectral-Domain OCT Measurements in Alzheimer’s Disease: A Systematic Review and Meta-analysis. Ophthalmology.

[2]          den Haan J, Verbraak FD, Visser PJ, Bouwman FH (2017) Retinal thickness in Alzheimer's disease: A systematic review and meta-analysis. Alzheimer's & Dementia : Diagnosis, Assessment & Disease Monitoring 6, 162-170.

[3]          Gupta VB, Chitranshi N, Haan Jd, Mirzaei M, You Y, Lim JKH, Basavarajappa D, Godinez A, Di Angelantonio S, Sachdev P, Salekdeh GH, Bouwman F, Graham S, Gupta V (2020) Retinal changes in Alzheimer's disease— integrated prospects of imaging, functional and molecular advances. Progress in Retinal and Eye Research, 100899.

[4]          Wu SZ, Masurkar AV, Balcer LJ (2020) Afferent and Efferent Visual Markers of Alzheimer’s Disease: A Review and Update in Early Stage Disease. Frontiers in Aging Neuroscience 12.

Author Response

Rev 2

The review article aims to explore the utility of retinal ganglion cell layer thickness as a potential biomarker for AD and its earlier stages.  It does so by reviewing studies that used OCT as a non-invasive eye imaging device to generate optical cross-section of the posterior pole of the eye that enables the segmentation of different layers of the retina in order to measure its thickness. There are several systemic reviews and meta-analysis have been published exploring the utility of OCT imaging in AD and MCI. Just to mention few:  

Recent meta-analysis also revealed the importance of GCL-IPL and GCC in AD and well discussed the issues that this manuscript highlighted and went well beyond that [1].  Another Meta-analysis highlighted that although there are retinal changes in AD, it might be too small to translate finding into clinic [2].

Further reviews:

Major review: Retinal changes in Alzheimer's disease— integrated prospects of imaging, functional and molecular advances[3]

Review: Afferent and Efferent Visual Markers of Alzheimer’s Disease: A Review and Update in Early Stage Disease[4]

The manuscript states that the major focus of the review will be the ganglion cell layer but then it talks about all the other retinal layers too. It doesn’t read well and feels chaotic. The results and the discussion section are not separated.

There is no novel information in this manuscript. Unfortunately, I do not see how the field would benefit from this manuscript.

References:  

[1]          Chan VTT, Sun Z, Tang S, Chen LJ, Wong A, Tham CC, Wong TY, Chen C, Ikram MK, Whitson HE, Lad EM, Mok VCT, Cheung CY (2018) Spectral-Domain OCT Measurements in Alzheimer’s Disease: A Systematic Review and Meta-analysis. Ophthalmology.

[2]          den Haan J, Verbraak FD, Visser PJ, Bouwman FH () Retinal thickness in Alzheimer's disease: A systematic review and meta-analysis. Alzheimer's & Dementia : Diagnosis, Assessment & Disease Monitoring 6, 162-170.

[3]          Gupta VB, Chitranshi N, Haan Jd, Mirzaei M, You Y, Lim JKH, Basavarajappa D, Godinez A, Di Angelantonio S, Sachdev P, Salekdeh GH, Bouwman F, Graham S, Gupta V (2020) Retinal changes in Alzheimer's disease— integrated prospects of imaging, functional and molecular advances. Progress in Retinal and Eye Research, 100899.

[4]          Wu SZ, Masurkar AV, Balcer LJ (2020) Afferent and Efferent Visual Markers of Alzheimer’s Disease: A Review and Update in Early Stage Disease. Frontiers in Aging Neuroscience 12.

We agree with the reviewer so we have added and numbered the following references:

  1. Gupta VB, Chitranshi N, Haan JD, Mirzaei M, You Y, et al. Retinal changes in Alzheimer's disease-integrated prospects of imaging, functional and molecular advances. Prog Retin Eye Res. 2020;2:100899. 
  2. Chan VTT, Sun Z, Tang S, Chen LJ, Wong A, Tham CC, et al. Spectral-Domain OCT Measurements in Alzheimer's Disease: A Systematic Review and Meta-analysis. Ophthalmology. 2019;126:497-510. 
  3. den Haan J, Verbraak FD, Visser PJ, Bouwman FH. Retinal thickness in Alzheimer's disease: A systematic review and meta-analysis. Alzheimers Dement (Amst). 2017;6:162-170.
  4. Wu SZ, Masurkar AV, Balcer LJ. Afferent and Efferent Visual Markers of Alzheimer’s Disease: A Review and Update in Early Stage Disease. Frontiers in Aging Neuroscience 2020

We believe that, although this manuscript do not bring any new information, herein we resume all recent studies in a easily way to better understand new findings in this field.

Reviewer 3 Report

This manuscript reviews studies that have used  optical retinal imaging to assess the integrity of retinal ganglion cells in patients with a clinical diagnosis of either Alzheimer’s disease or mild cognitive impairment.

It is a useful contribution, but has a number of shortcomings:

  1. The authors use the term ‘AD patients’, which is not appropriate since a histological diagnosis has not been carried out in all cases. Therefore, the term ‘probable AD patients’ would be better.
  2. It would be useful to describe the criteria by which a diagnosis of MCI was obtained and whether this diagnosis was consistent across the various studies.
  3. The table summarising the various studies would be greatly improved if the magnitude of the differences (possibly expressed as % of control) described was included, along with the relevant P values.
  4. What seems to be missing from the literature, is longitudinal studies of the integrity of ganglion cells in patients with probable AD. This should be considered and discussed.
  5. The authors should also consider, as much as they can, the sensitivity and selectivity of ganglion cell thinning as a biomarker of Alzheimer’s disease.
  6. Some discussion of the pathology within the eye in relation to pathology across the brain would also be helpful.

Author Response

This manuscript reviews studies that have used optical retinal imaging to assess the integrity of retinal ganglion cells in patients with a clinical diagnosis of either Alzheimer’s disease or mild cognitive impairment.

It is a useful contribution, but has a number of shortcomings:

  1. The authors use the term ‘AD patients’, which is not appropriate since a histological diagnosis has not been carried out in all cases. Therefore, the term ‘probable AD patients’ would be better.

We agree so we have changed the term all across the manuscript. 

  1. It would be useful to describe the criteria by which a diagnosis of MCI was obtained and whether this diagnosis was consistent across the various studies.

We agree so we have added the following:

“Besides, definition of MCI was used metting research diagnostic criteria for probable AD MCI or with evidence of the AD pathophysiological process (in most cases defined by a positive amyloid-PET) following the recommendations of the National Institute on Aging-Alzheimer’s Association (70), but criteria should be consistent.”

  1. McKhanna GM, Knopmanc DS, Chertkowd H, Hymanf BT, Jack Jr CR, Kawas CH, et al. The diagnosis of dementia due to Alzheimer’s disease: Recommendations from the National Institute on Aging- Alzheimer’s Association workgroups on diagnostic guidelines for Alzheimer’s disease. Alzheimers Dement. 2011;7: 263–269.
  2. The table summarising the various studies would be greatly improved if the magnitude of the differences (possibly expressed as % of control) described was included, along with the relevant P values.

We added % of difference and P values all along the table.

  1. What seems to be missing from the literature, is longitudinal studies of the integrity of ganglion cells in patients with probable AD. This should be considered and discussed.

We agree, so we added the following statement: “However, today might be soon to consider optic nerve or retinal biomarkers as a reliable biomarkers of AD, as a limited number of studies have compared OCT retinal measures with neuroimaging biomarkers and very few longitudinal within-subject studies of retinal structural changes in AD are published.”

  1. The authors should also consider, as much as they can, the sensitivity and selectivity of ganglion cell thinning as a biomarker of Alzheimer’s disease.

We added the values of AUC curves of a manuscript, but the conclusion statement resumes our opinion of this point.

  1. Some discussion of the pathology within the eye in relation to pathology across the brain would also be helpful.

We added this paragraph in discussion section:

“There might be two mechanisms which explain GCL damage. The first proposed that cerebral pathologic features of AD may affect the visual pathway and cause retrograde degeneration of the optic nerve (29), and subsequently damage of the GCL, because AD pathologic features can be found in subcortical visual centers, including the lateral geniculate nucleus and superior colliculus (30). In agreement with this hypothesis, GC abnormalities also were associated with non-AD dementias (31-33), strokes (34,35) and other neurodegenerative diseases including multiple sclerosis (36-38), neuromyelitis optica (37), and cerebral atrophy (39). Alternatively, it is also possible that AD pathologic features occur simultaneously both in the brain and the retina, leading to thinning of the retinal neuronal layers. Aβ pathologic characteristics, including Aβplaques and specific signs of neuroinflammation, have been identified in ocular tissues of both probable AD patients (24,40) and animal models of AD (27, 41-45).”

Round 2

Reviewer 3 Report

The authors have addressed my concerns.

One minor point - in Table 1, it would be good to more clearly indicat what '%' means in  the sentence 'Significant results are shiowed as (%, P)'